# Transmission Line Defect Target-Detection Method Based on GR-YOLOv8

**DOI:** 10.3390/s24216838

**Published:** 2024-10-24

**Authors:** Shuai Hao, Kang Ren, Jiahao Li, Xu Ma

**Affiliations:** 1College of Electrical and Control Engineering, Xi’an University of Science and Technology, Xi’an 710054, China; haoxust@163.com (S.H.); kk2580256@163.com (K.R.); 18406060425@stu.xust.edu.cn (J.L.); 2Huaneng Longdong Energy Co., Ltd., Qingyang 744500, China

**Keywords:** transmission line defect detection, vision transformer, YOLOv8, light weight, loss function

## Abstract

In view of the low levels of speed and precision associated with fault detection in transmission lines using traditional algorithms due to resource constraints, a transmission line fault target-detection method for YOLOv8 (You Only Look Once version 8) based on the Rep (Representational Pyramid) Visual Transformer and incorporating an ultra-lightweight module is proposed. First, the YOLOv8 detection network was built. In order to address the needs of feature redundancy and high levels of network computation, the Rep Visual Transformer module was introduced in the Neck part to integrate the pixel information associated with the entire image through its multi-head self-attention and enable the model to learn more global image features, thereby improving the computational speed of the model; then, a lightweight GSConv (Grouped and Separated Convolution, a combination of grouped convolution and separated convolution) convolution module was added to the Backbone and Neck to share computing resources among channels and reduce computing time and memory consumption, by which the computational cost and detection performance of the detection network were balanced, while the model remained lightweight and maintained its high precision. Secondly, the loss function Wise-IoU (Intelligent IOU) was introduced as the Bounding-Box Regression (BBR) loss function to optimize the predicted bounding boxes in these grid cells and shift them closer to the real target location, which reduced the harmful gradients caused by low-quality examples and further improved the detection precision of the algorithm. Finally, the algorithm was verified using a data set of 3500 images compiled by a power-supply inspection department over the past four years. The experimental results show that, compared with the seven classic and improved algorithms, the recall rate and average precision of the proposed algorithm were improved by 0.058 and 0.053, respectively, compared with the original YOLOv8 detection network; the floating-point operations per second decreased by 2.3; and the picture detection speed was increased to 114.9 FPS.

## 1. Introduction

In recent years, owing to rapid economic development, the total length of high-voltage transmission lines worldwide has exceeded four million kilometers. In order to ensure the safety and quality of transmission, power line inspections have received increasingly more attention throughout the world [1]. Due to their long-term exposure to harsh external environments such as high winds and heavy rains, there has been high frequency of faults in transmission lines [2,3,4]. Therefore, regular transmission line inspections are very important to ensuring their safety and stability. UAV inspection has been widely used in power line inspections because of its flexibility, high speed, and safety. However, the massive amount of inspection data cannot be timely processed due to limited equipment resources, etc. Consequently, many detection algorithms are not readily usable, lacking the required speed and precision, which causes a large workload [5]. Therefore, given the limited nature of the equipment resources, achieving accurate and efficient defect defection from the image data set generated in UAV inspection is of great practical significance.

At present, algorithms based on deep learning theory have made significant progress in the field of target detection. This approach is mainly based on a data-driven method for target detection. It has the advantages of high detection accuracy, strong robustness, and multi-target detection capabilities. This is a research hotspot in computer vision. The relevant deep learning algorithms can be roughly divided into two categories: One is the two-stage algorithm, which is represented by R-CNN [6] (Region-based Convolutional Neural Networks), Fast R-CNN [7], and FPN [8] (Feature Pyramid Network). Wang et al. [4] used the R-CNN algorithm to locate the spacers, shock absorbers, and isobaric rings on the UAV inspection images, and realized the real-time and accurate identification of these transmission line components. Liang et al. [9] proposed a transmission line defect detection method based on deep learning, using Faster R-CNN to construct a transfer learning detection model. This method has good robustness for defect detection under different illumination conditions. However, the two-stage algorithm is often computationally complex, as characterized by its detailed candidate region generation and classification process detection, resulting in its slow speed and difficulty in meeting the detection requirements involved with massive data.

The other category comprises a one-stage algorithm, including the SSD [10] (Single Shot MultiBox Detector) and YOLO [11] (You Only Look Once) series, as well as RetinaNet [12], EfficientDet [13], and other algorithms. These algorithms have obvious advantages in speed and are more suitable for large-scale detection tasks that require high levels of efficiency. By extracting high-quality features from aerial images and using multi-level perception, the advantages characteristic of global and local information can achieve a favorable balance and effectively improve the detection accuracy [14]. The method designed by Hao et al. [15] improves the feature extraction ability of the network and performs effective feature fusion. However, the applicability of the detection algorithm to the inspection of transmission lines is not considered in the above research, and graphics cards and processors associated with the transmission line inspection equipment are seldom timely updated. As a result, the precision and speed of many algorithms are not achieved because of these inadequate software and hardware configurations, resulting in the failure of the timely processing of massive amount of defect-related image data generated in inspections.

In view of the above problems, a YOLOv8 multi-fault target-detection method for transmission lines is proposed based on Visual Transformers and ultra-lightweight modules, one which is simply referred to as GR-YOLOv8. The main innovations and contributions of this paper are as follows:

(1) As to the obvious high requirements inherent in the network computing and feature map redundancy of the YOLOv8 algorithm, a Rep Visual Transformer module is introduced in the Neck part, and the multi-head self-attention module is used to enhance the model’s ability to learn image features and reduce the feature dimensions to be processed and the computational burden, thereby improving the computational speed of the model.

(2) Give the generally limited equipment resources and the low applicability of the detection algorithm, the lightweight GSConv convolution module was introduced in the Backbone part; in this module, an optimization strategy for the combination of grouped convolution and separated convolution is adopted to reduce the amount of computation and reduce memory loss. Computing resources can also be dynamically allocated based on actual hardware performance to balance computing costs and detection performance, which not only achieves higher detection precision, but also reduces power and energy consumption. Especially in terms of small-size feature maps, GSConv convolution creates a lower computational burden than does standard convolution, which improves the precision as to small targets to certain extent.

(3) The loss function CloU was modified to Wise-IoU. After considering the influence of the position of the bounding box, the tolerance of the model for position errors was adjusted to optimize the final test result. In addition to the reduction in the competitiveness of high-quality anchor frames, the harmful gradients generated by low-quality examples were also reduced. Therefore, the bounding-box regression was more accurate, which further improves the detection precision of the model.

## 2. Principle of YOLOv8 Detection Algorithm

The YOLOv8 network architecture includes three key parts (as shown in Figure 1): the Backbone network for feature extraction, the Backbone network for feature fusion, and the detection head for the final identification and detection.

The Backbone network is a local cross-stage network designed to reduce the computational load and optimize gradient propagation. In order to capture spatial information more efficiently, it integrates the spatial pyramid pooling module [16]. Compared to YOLOv5’s PAN-FPN, the Neck network eliminates the convolution of the upsampling stage. Instead, downsampling is performed prior to the upsampling. At the same time, the C3 model is replaced with a lighter C2f module [17], which enhances the model’s adaptability as to targets of different sizes and shapes. The detection head utilizes the currently popular decoupling structure, which not only reduces the number of parameters and computational complexity, but also improves the generalization and stability of the model.

The loss of the bounding box of the YOLOv8 algorithm is calculated by the Conv2d layer. At the same time, the traditional anchor-based prediction method is abandoned, and replaced with an anchor-free strategy aiming to directly predict the center coordinates and aspect ratio of the target, thereby minimizing anchor boxes and improving the detection speed and precision.

## 3. GR Lightweight Model

Classic lightweight models include MobileNets [18,19], ShuffleNets [20,21,22], and GhostNet [23,24,25]. As one example, the core feature of MobileNets is the use of a special Depthwise Separate Convolution (DSC), in which the traditional convolution is divided into two independent functions of depthwise convolution and pointwise convolution. The depthwise convolution is very efficient in processing high-resolution images, while pointwise convolution involves the application of 1 × 1 convolution on each channel separately, a process which actually consists of the global-average pooling of all channels before the expansion is expanded. Therefore, the effect of pointwise convolution is similar to that of the feature fusion of channels. However, 1 × 1 intensive convolution takes up more computing resources, and often leads to lower levels of detection precision. In addition, many lightweight models use only DSC in their basic architecture design, from the beginning to the end, which causes the direct amplification of the defects of DSC in the Backbone network. Therefore, precision has been significantly reduced, regardless of whether it is used for image classification or detection.

In order to make the output of DSC close to scattering (SC), a new method was introduced with the use of a mixed convolution of SC, DSC, and shuffle; this was called Grouped Spatial Convolution 9 GSConv [26]. This convolution preserves these connections as much as possible with lower time complexity, especially in mobile devices or circumstances where resources are limited. Therefore, GSConv reduced the amount of computing and memory loss, and improved the detection of small targets. The time complexity formulas of SC, DSC, and GSConv are as follows:(1)$${\mathrm{Time}}_{\mathrm{SC}}∼0\left(W\cdot H\cdot K_{1}\cdot K_{2}\cdot C_{1}\cdot C_{2}\right)$$
(2)$${\mathrm{Time}}_{\mathrm{DSC}}∼0\left(W\cdot H\cdot K_{1}\cdot K_{2}\cdot 1\cdot C_{2}\right)$$
(3)$${\mathrm{Time}}_{{\mathrm{GSC}}_{\mathrm{onv}}}∼0\left(W\cdot H\cdot K_{1}\cdot K_{2}\cdot \frac{C_{2}}{2}\cdot \left(C_{1}+1\right)\right)$$
where Time is time complexity; W is the width of the output feature map; H is the height of the output feature map; K1·K2 is the size of the convolution kernel; C1 is the number of channels in each convolution kernel, as well as the number of channels in the input feature map; and C2 is the number of channels in the output feature map. The structure of the GSConv model is shown in Figure 2.

Shuffle is used for the penetration of the SC information into the various parts of the DSC information. This method allows SC information to be evenly exchanged with local characteristic information on different channels and completely mixed into the DSC output without cumbersome steps, which not only optimizes the model performance and reduces computational complexity and memory consumption but also maintains a certain degree of model expression, especially in mobile devices or circumstances where resources are limited.

In addition, RepViT [27] utilizes an improved Transformer architecture, one which combines traditional convolutional networks with Visual Transformer (ViT). In particular, it introduces a multi-head self-attention module to enhance the model’s ability to learn image features. The multi-head self-attention mechanism allows the model to process information associated with different scales and locations in parallel. The overall architecture of the RepViT model is obtained through 14 optimization steps, such as the alignment of the training formula, delay of metric, the alignment of the training strategy, the optimization of the block design, and the separation of the token mixer. The structure is shown in Figure 3.

In this architecture, H and W are the width and height of the input image; Ci is the size of the i-th channel; and B is the size of the batch.

In the end, the Rep Visual Transformer module was introduced in the Neck part, and the lightweight GSConv convolution module was added to the Backbone and Neck. The lightweight design allows the model to reduce computing time and memory consumption without excessively sacrificing precision and is suitable for real-time applications or object detection on a resource-limited device.

## 4. Wise-IoU Loss Function

The bounding-box loss function plays a vital role in improving the model’s target-detection speed. Since current research focuses on improving the matching of precise bounding boxes, the over-processing of many common poor-quality samples in the actual training data may adversely affect the detection effect.

The original YOLOv8 uses GIOU [28] as the loss function of the bounding box. Although it solves the failure to predict the distance between the intersection over union (IOU) prediction box and the real box when they do not intersect, GIOU cannot measure their relative positional relationship when the prediction box and the real box are in an inclusive relationship. As a result, the detection effect of the model is affected.

Therefore, the Wise-IoU [29] loss function is introduced herein. In this function, with the adoption of the two core strategies of Feature Pyramid Networks (FPN) and finer regional proposal generation technology, low-level feature maps were used for detection of small objects while high-level feature maps were used for detection of large objects, which allows the capture of more potential target information. At the same time, the loss function evaluates the quality of the anchor frame based on its outlier. By identification of the small offset of the real bounding box of the predicted object relative to the preset candidate area, the prediction box can cover the target object more accurately, and reduce the negative gradient effect of low-quality samples, without overemphasizing the high-quality anchor frame. Therefore, Wise-IoU has the significant ability to process medium-quality anchor frames, which improves the overall performance of the detector. The calculation formula for Wise-IoU is:(4)$$\begin{array}{rrr} L_{WIoU} & = & R_{WIoU}\times L_{IoU}L_{WIoU\nu 1}=R_{WIoU}\times L_{IoU} \end{array}$$
(5)$$R_{WIoU}=exp\frac{{\left(b_{c_{x}}^{g^{t}}-b_{c_{x}}\right)}^{2}+{\left(b_{c_{y}}^{g^{t}}-b_{c_{y}}\right)}^{2}}{c_{w}^{2}+c_{h}^{2}}$$
(6)$$L_{IoU}=1-IoU$$

The parameters of Equations (4)–(6) are shown in Figure 4. ${{(b}_{c_{x}}^{g^{t}}-b_{c_{x}})}^{2}$ is the Euclidean distance between the center points of the real box and the prediction box; h and w are the height and width; and $c_{h}$ and $c_{w}$ are the height and width of the smallest closed box composed of the prediction box and the real box.

## 5. Experiments and Analysis of Results

Innovative improvements were made to the target-detection algorithm of the YOLOv8 model, including the application of Rep Visual Transformer and lightweight GSConv module, and the optimization of the loss function into Wise-IoU, etc. The structural diagram of the proposed GR-YOLOv8 network is shown in Figure 5.

In order to simulate an environment with limited on-site equipment resources, the most common on-site hardware and software configurations were adopted: Windows 10 operating system, NVIDIA GeForce GTX1050 graphics card (with 4 GB video memory), Intel (R) Core (TM) i5-8300H, Pytorch 3.8, and PyCharm 2021.2.4.

### 5.1. Collection and Pre-Processing of Experimental Data

The image data associated with 330 kV high-voltage transmission lines and acquired by inspections using drones conducted by the on-site department of a power supply administration agency over the past 4 years were used. The 12 common problems with transmission lines included wire damage, insulator shedding, bird nest formation and spacer rod disconnection. Some defect targets are shown in Figure 6.

A data set of 3500 inspection images from a power supply administration agency was selected, and their defects were marked in detail with LabeIImg. The classifications and numbers of the defects are shown in Table 1.

### 5.2. Model Training

The various anomalies in the data set were randomly divided into training sets and test sets at proportions of 70% and 30%, respectively. During the model training, the size of all image samples was adjusted to 640 × 640 to ensure their consistency and reduce the computational burden caused by the original size difference. The stochastic gradient descent optimization algorithm was used to update the model parameters, and the total number of iterations was set to 100 times with a batch size of 16, which is suitable for scenarios in which a quick response to data streams is expected with limited resources. The initial learning rate was set to 0.01. The learning rate determines the update speed of the model parameters. If it is set too high, it may lead to unstable training, and if it is set too low, it may lead to slow convergence. The momentum used was 0.937. This optimization strategy not only considers the direction of the current gradient in each iteration, but also retains a part of the memory of the previous gradient direction, which helps to smooth the update and reduce oscillations, which is especially significant in dealing with considerable local minima.

### 5.3. Model Evaluative Indicators

The evaluative indicators of the model were precision (P), recall (R), floating-points (Flops), average precision (AP), mean average precision (mAP), and frame rate per second (FPS). The calculation formulas associated with some of the evaluative indicators are:(7)$$P=\frac{N_{TP}}{N_{TP}+N_{FP}}\times 100\%$$
(8)$$R=\frac{N_{TP}}{N_{TP}+N_{FN}}\times 100\%$$
(9)$$AP=\int_{0}^{1}P\left(R\right)dR$$
(10)$$mAP=\frac{1}{n}\sum_{i=1}^{n}AP_{i}$$
(11)$$FPS=\frac{1}{t}$$ P is the proportion of positive examples among those predicted to be positive by the model; R is to the proportion of actual positive examples correctly recognized by the model; $N_{TP}$ is the positive sample accurately predicted by the algorithm; $N_{FP}$ is the positive sample misjudged by the algorithm; $N_{FN}$ is the positive sample not predicted by the algorithm; Flops is the number of floating-points required during the operation of the model, which is used to measure the computational complexity and hardware resource consumption of the model; the average precision (AP) is determined by calculating the area under the P–R curve, and the higher the value, the better the performance of the model in identifying a specific target; mAP is the average of the APs of all categories; and n is the total number of categories. For the i-th category of targets, the larger the mAP, the higher the precision of the algorithm. FPS is used to measure the detection speed of the algorithm, which refers to the average time it takes to detect a frame of image. The larger the FPS, the higher the detection speed and the better the real-time performance.

### 5.4. Analysis of Results

The real-time change curve of the loss function during the model training is shown in Figure 7.

Figure 7 shows the comparison of the training processes of the GR-YOLOv8 and YOLOv8 algorithms. In the initial stage, the loss values of the two are close, but in subsequent iterations, the loss value of the proposed algorithm drops faster and more steadily, and is always at a low level. In the end, the loss value of GR-YOLOv8 was approximately 0.9, while the value for YOLOv8 was approximately 1.9. From the experimental results in Figure 6, it can be seen that GR-YOLOv8 algorithm achieves better training results.

The mAP@0.5 curves of the GR-YOLOv8 and YOLOv8 algorithms are shown in Figure 8.

It can be seen from Figure 8 that after 20 cycles of training iterations, the mAP@0.5 of GR-YOLOv8 algorithm rose to about 0.9. Comparatively, after 70 cycles of training iterations, the mAP@0.5 of YOLOv8 rose to only about 0.9, and this level was not sustained for long. In the end, the mAP@0.5 of GR-YOLOv8 stabilized at about 0.935, and that of YOLOv8 stabilized at about 0.882. Therefore, compared with the YOLOv8 algorithm, the proposed algorithm is improved to a certain extent in terms of the mean of average precision.

From the test results in Table 2, it can be seen that the average precision of the proposed GR-YOLOv8 algorithm reached 0.935 and the recall rate reached 0.922.

In order to verify the impacts of the use of Gsconv, RepViT, and Wise-IoU on the performance of the model, ablation experiments were conducted based on the original yolov8 algorithm, as shown in Table 3.

In Table 3, “-” means “not adopted”, and “√” means “adopted”. In the first line, the mAP of the original YOLOv8 algorithm was 0.872. After improvements in three aspects, the number of parameters decreased by 889,846, and the FPS was greatly improved. The mAP reached 0.935, which was 0.053 higher than that of the original YOLOv8 algorithm. It can be seen that the lightweight design of GR-YOLOv8 algorithm model achieves a very obvious effect, and the detection precision has also been improved accordingly. In order to further objectively evaluate the advantages of the GR-YOLOv8 algorithm, some classic and improved algorithms were compared through experiments. All algorithms were trained using the above data sets and configurations. The results are shown in Table 4.

As shown in the comparative experimental results in Table 4, the floating-points, recall rate, mean of average precision, and detection rate of YOLOv8 were 9.2, 0.864, 0.882, and 68.7 FPS, respectively. Then, the EMA [30,31] (Efficient Multi-Scale Attention in Cross-Space) and simEMA (Simplified Exponential Moving Average Attention) attention mechanisms were tried, for which the core C2 module was replaced with C2f-EMA and C2f-simEMA. Although this change slightly improves the image detection speed and floating-point computing speed, after the introduction of the EMA module, the recall rate was reduced by 0.012 compared to YOLOv8. After the application of the simEMA module, the recall rate was reduced by 0.021, and the average precision was also significantly reduced by 0.033.

As to the Swin Transformer algorithm, its architecture design is similar to that of the traditional convolution hierarchy, in which the resolution of each layer is halved but the number of channels is doubled. The principle is to divide the image into windows, that is, the division of input sequence into windows of fixed size or dynamic size. Then, a standard self-attention mechanism is used to calculate local features inside each window, followed by interaction and spicing of these windows. The results show that compared with YOLOv8, although the algorithm had improved the recall rate by 0.044 and the mean of average precision by 0.035, the number of floating-point operations increased significantly, and the image detection speed also decreased by 4.6 FPS.

Dynamic Snake Convolution (DSConv) adopts the precise segmentation of a tubular structure to ensure the precision and efficiency of downstream tasks. Hence, it was tried in the application of defect target-detection for transmission lines. However, because most defects are at the towers and poles of transmission lines, the experimental results were too unsatisfactory in terms of floating-point calculation, recall rate, mean of average precision, and image detection speed. After the addition of the detection head module, the ability of detection as to small targets was greatly improved, and the recall rate and average precision were increased by 0.048 and 0.043, respectively, although the image detection speed decreased significantly.

The detection performance of the GR-YOLOv8 algorithm was significantly improved. The recall rate and the mean of average precision increased by 0.058 and 0.053, respectively. The floating-point operations per second decreased by 2.3, and the image detection speed increased to 114.9 FPS. With its lightweight design and ease of on-site application in an environment with resource constraints, the detection precision is improved while the computational complexity and memory consumption are reduced, which verifies the feasibility of the improvement plan.

In order to verify the advantages of the proposed algorithm as to detection, four groups of scenes were selected for the testing, including those with multiple categories, partial occlusion, small scale, and complex backgrounds. In Figure 9a, these are arranged from left to right. The defects in the images were marked with rectangular boxes. The comparison results for YOLOv8, EMA, Swin Transformer, DSConv, detection head, Gsconv, and GR-YOLOv8 detection algorithms in the experiments are shown in Figure 9.

As shown in Figure 9, the defects of the transmission lines were clearly identified, and the detection confidence of YOLOv8 was between 0.7 to 1.0, while values for other algorithms fluctuated between 0.3 and 1.0. It is particularly obvious that the Depthwise Separate Convolution (DSConv) failed to identify the defects in some cases. In contrast, the GR-YOLOv8 algorithm exhibits a high level of confidence, which ensures that defects are precisely identified in most cases. In addition, the algorithm in this paper can detect a variety of defects of transmission lines in a single image, which shows satisfactory adaptability, stability, and robustness. The analysis of the four sets of test results is conducted in the following paragraphs (Group 1 is on the left).

There are many types of defects in the Group 1 experiments, and they are largely different in shape and size. The Dynamic Snake Convolution (DSConv) performed poorly in dealing with multiple types of defects and failed to achieve the necessary precision. Other algorithms have a high degree of responsiveness in the detection of multiple types of defects, and without obvious missed detections. Among them, the detection results of the YOLOv8, Swin Transformer, detection head, and Grouped Spatial Convolution (Gsconv) algorithms were consistent. The confidence for the detection results of Exponential Moving Average (EMA) was low, and the GR-YOLOv8 algorithm can more precisely identify and locate multiple types of defects, achieving better detection results.

The Group 2 experiments involve the detection of defects in complex environments. The experimental results show that YOLOv8, EMA, Swin Transformer, and DSConv all missed detections when dealing with complex backgrounds, which indicates that these algorithms are not responsive to defects against complex backgrounds. The GR-YOLOv8 detection algorithm had a high response rate in this experiment, and the confidence of the detection results was also high, 0.7 and 0.8, respectively, which indicates better performance in complex environments.

The Group 3 experiments involve the detection of insulator defects, which is critically important in the defect inspection of a power system. The experimental results show that the detection results for the Depthwise Separate Convolution (DSConv) were low, with a confidence level of only 0.4, and its ability in feature extraction and the handling of specific types of defects is limited to certain extent, a limitation which requires further optimization and improvement as to the detection of insulator defects. The confidence of the other six algorithms in the detection of insulator defects was 1, which shows that they achieve higher precision and reliability and better detection performance.

Group 4 experiments involve the detection of small defects. The DSConv algorithm exhibits obvious false-negative cases, which demonstrates its obvious disadvantages in this respect. The confidence of the detection results for the EMA and Swin Transformer algorithms was only 0.3, and that of the YOLOv8 algorithm was 0.9. Comparatively, the confidence of the detection results of GR-YOLOv8 algorithm was 1.0. Therefore, it can precisely identify and locate small defects, showing better detection results.

## 6. Conclusions

In view of the present low levels of speed and precision in the detection of defects in transmission lines with traditional algorithms and given situations in which resources are limited, a YOLOv8 target fault-detection method for transmission lines is proposed, based on Visual Transformers and ultra-lightweight modules. The Rep Visual Transformer module was introduced in the Neck part to integrate the pixel information for the entire image through its multi-head self-attention. This enabled the model to learn more global image features, which improved the computational speed of the model and thus solved for feature redundancy and high network computing demands; the lightweight GSConv convolution module was added to the Backbone and Neck to share computing resources among channels and reduce computing time and memory consumption, and the computational cost and detection performance of the detection network were thereby balanced to improve the detection precision and speed, especially in the detection of small targets. At the same time, the loss function Wise-IoU was introduced to reduce the harmful effects caused by low-quality examples, and further improved the detection precision of the algorithm. Compared with seven classic or improved high-speed algorithms such as EMA, Swin Transformer, and Detection head, GR-YOLOv8 can reduce the number of parameters and calculations, as well as the memory usage, in circumstances involving limited software and hardware resources, which reduces computing time and energy consumption while maintaining a sufficient degree of feature extraction. This balances computing costs and detection performance in commonly used equipment, greatly improving the detection speed while improving the precision in defect detection to certain extent.

## Figures and Tables

**Figure 1 sensors-24-06838-f001:**
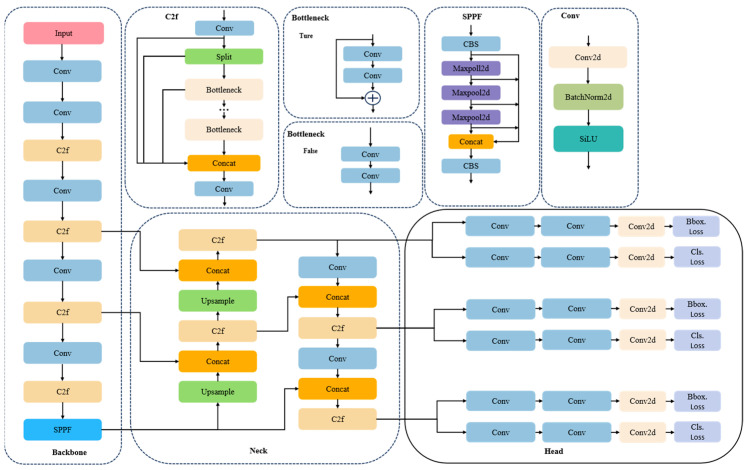
YOLOv8 network structure.

**Figure 2 sensors-24-06838-f002:**
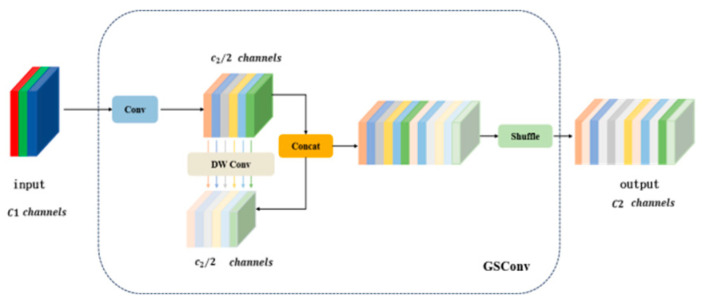
GSConv network structure.

**Figure 3 sensors-24-06838-f003:**
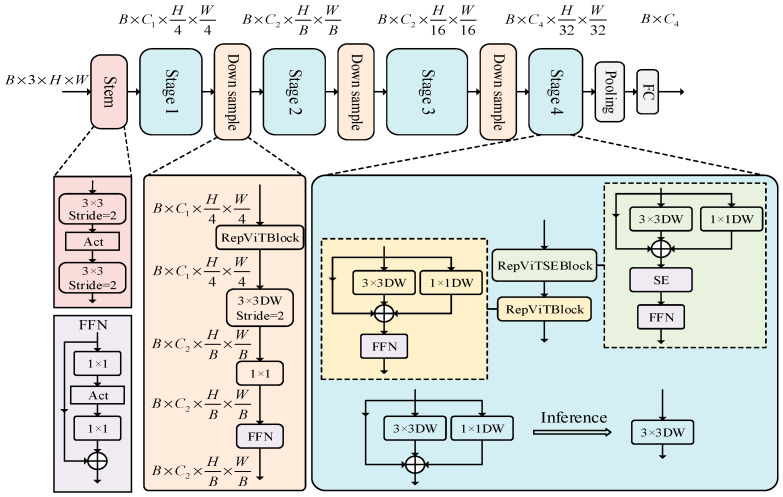
The overall architecture of the RepViT model.

**Figure 4 sensors-24-06838-f004:**
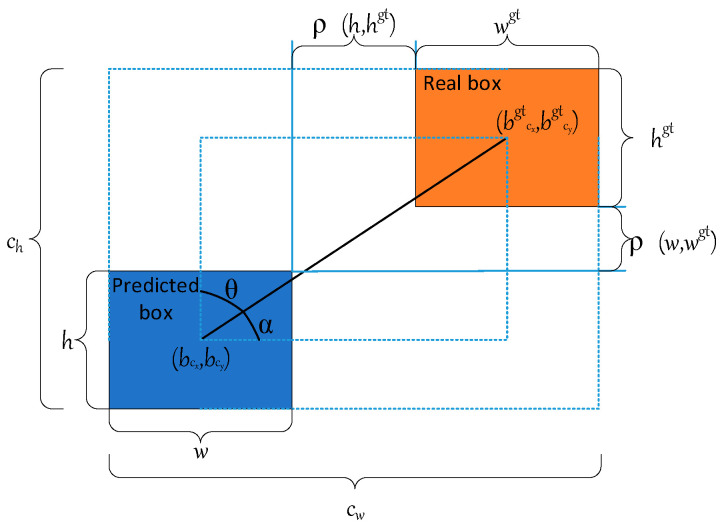
Schematic diagram of the parameters of the loss function.

**Figure 5 sensors-24-06838-f005:**
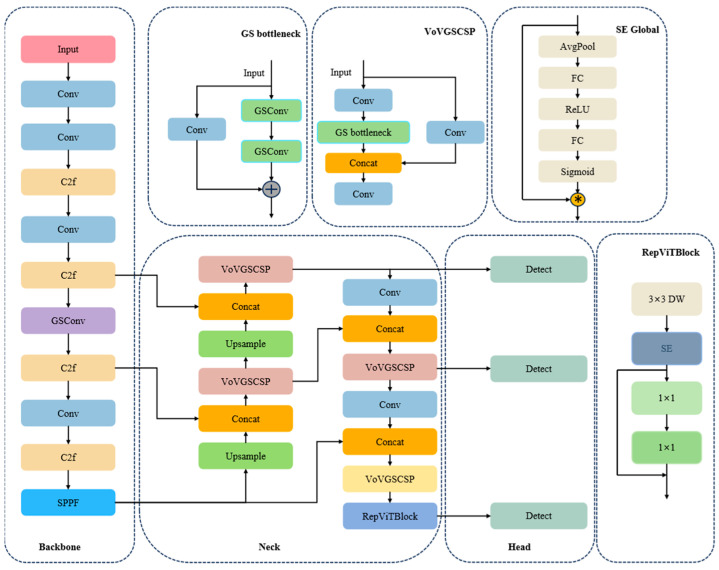
GR-YOLOv8 network structure diagram.

**Figure 6 sensors-24-06838-f006:**
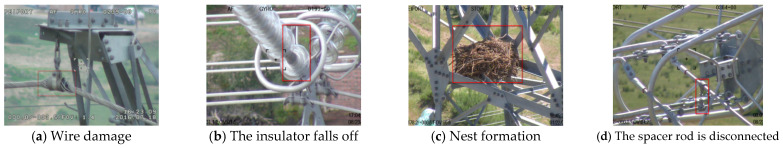
Common defects in transmission lines.

**Figure 7 sensors-24-06838-f007:**
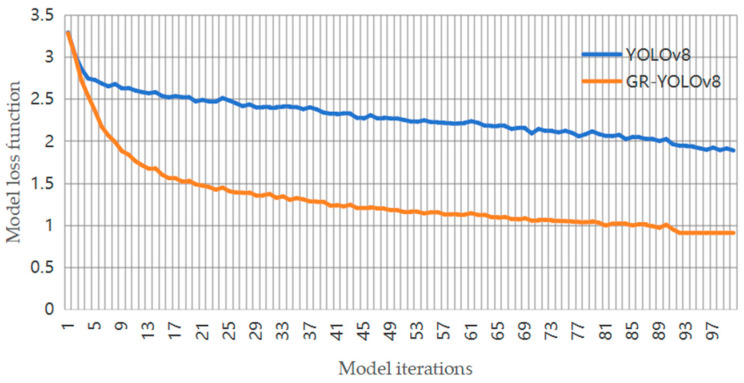
Loss function curve.

**Figure 8 sensors-24-06838-f008:**
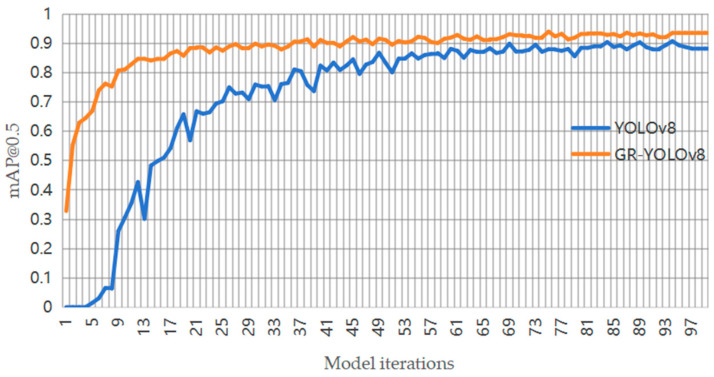
mAP@0.5 contrast curve.

**Figure 9 sensors-24-06838-f009:**
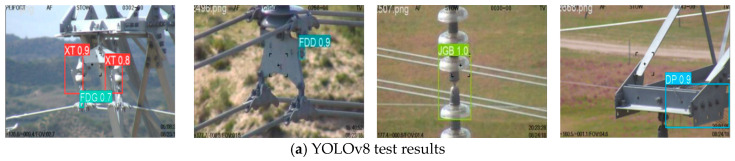
Comparison of detection results.

**Table 1 sensors-24-06838-t001:** Defect name and quantity.

Fault Type	Custom Name	Quantity
Pin having come off	XT	342
Wire damage	DXS	316
Nut having come off	LST	347
Discharge gap too large	FDG	310
Insulator sheds	JYT	212
Bird’s nest	NC	235
Gap design problems	FDSJ	262
Anti-bird spur damage	NS	231
Gap shorting	FDD	310
Gaskets missing	DP	323
Spacer bar disconnected	JGB	364
Sundry issues	ZW	248

**Table 2 sensors-24-06838-t002:** Detection results for the GR-YOLOv8 model on various targets.

Defect Type	Precision	Recall
XT	0.955	1
DXS	0.974	0.939
LST	0.976	0.970
FDG	0.995	1
JYT	0.953	0.943
NC	0.969	0.964
FDSJ	0.995	1
NS	0.643	0.500
FDD	0.911	0.957
DP	0.948	0.914
JGB	0.956	0.939
ZW	0.946	0.939
all	0.935	0.922

**Table 3 sensors-24-06838-t003:** Ablation experiments.

Gsconv	RepViT	Wise-IoU	mAP	FPS	Parameter Quantity
-	-	-	0.882	68.7	3,740,727
√	-	-	0.894	90.3	3,185,239
√	√	-	0.907	114.9	2,850,881
√	√	√	0.935	114.9	2,850,881

**Table 4 sensors-24-06838-t004:** Performance comparisons of different algorithms.

Model	FLOPS	Recall	mAP	FPS
YOLOv8	9.2	0.864	0.882	68.7
EMA	8.1	0.852	0.889	74.1
simEMA	9.1	0.843	0.849	68.9
Swin Transformer	9.8	0.908	0.917	64.1
DSConv	21.9	0.085	0.077	52.1
Detection head	15.4	0.912	0.925	28.3
Gsconv	8.4	0.876	0.894	87.2
GR-YOLOv8	6.9	0.922	0.935	114.9

## Data Availability

Restrictions apply to the availability of these data. Data were obtained from power grid bureau and are available from the authors with the permission of power grid bureau.

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
