# Peer review of "Transmission Line Defect Target-Detection Method Based on GR-YOLOv8"

_sensors, 2024, doi:10.3390/s24216838_

Round 1
Reviewer 1 Report
Comments and Suggestions for Authors
The research topic is very interesting. However, I would like to suggest some revisions to improve the paper by addressing a few issues. Please see the proposed changes below.
1) The paper proposes a model called GR-YOLOv8, which improves the performance of the existing YOLOv8, to design an algorithm for detecting faults in transmission lines. The research topic is very interesting. However, the paper does not clearly describe the specific issues this model aims to address compared to existing algorithms. It would be better to provide a clearer explanation of the advantages of the proposed solution in specific situations and the shortcomings of the previous models.
2) The paper mentions that 'the slow detection speed and low accuracy of traditional transmission line fault detection algorithms' are problematic, but it is unclear in which specific situations these issues occur. For example, it would be helpful to explain concrete scenarios where existing algorithms suffer from low accuracy or slow speed, and the practical problems that arise as a result (e.g., inability to respond in real-time, limitations in processing large-scale data). This would help readers clearly understand the severity of the problem.
3) The paper focuses on the specific domain of transmission line fault detection, but it lacks an explanation of how GR-YOLOv8 can be applied in real-world scenarios. There is insufficient discussion on whether this model operates reliably under various weather, lighting, and environmental conditions, and what advantages it offers for actual transmission line inspections. For instance, it would be beneficial to emphasize its potential for real-time application or its advantages in inspection systems using UAVs (Unmanned Aerial Vehicles).
4) Relevant studies and literature are not sufficiently cited. For example, if there is a performance comparison with other YOLO variants or Transformer-based models, citing more related works could enhance the contribution of this paper.
Reviewer 2 Report
Comments and Suggestions for Authors
Multi-fault target detection method for transmission lines based on GR-YOLOv8, introduces Rep vision transformer module, GSConv lightweight convolutional module, and Wise-IoU loss function to YOLOv8, aiming at the problems of slow detection speed and low accuracy of traditional transmission line fault detection algorithms. My general comments on the manuscript are as follows.
1、The introduction section is not comprehensive enough, to supplement the shortcomings of existing transmission line fault detection algorithms.
2、The section of GR lightweight model, Wise-IoU loss function, and the Figure 4. GR-YOLOv8 Network structure diagram need to be further integrated.
3、The dataset needs to provide the number of fault images of various categories.
4、The experimental comparison results in Figure 7 should provide images with clear differentiation as much as possible.
5、All formulas need to be edited using a formula editor.
6、The number of references is relatively small.

NO
Reviewer 3 Report
Comments and Suggestions for Authors
1. Introduction
(1) The introduction part is not systematic enough to summarize the previous work, common single-stage algorithms such as RetinaNet, EfficientDet and other algorithms, two-stage detection algorithms such as FPN, Fast R-CNN, etc., and it is suggested to increase the summaries of related work.
(2) The author points out that other people's algorithms do not consider the compatibility of the speed and accuracy of the algorithms, but the author's description of the previous work says that other people's algorithms ultimately improve the generalization and speed, which is logically contradictory, and it is recommended that the descriptions of the relevant work be adjusted to highlight the difference between the innovative points of his work and others and the practical significance of his work.
(3) Both 1) and 3) of the main innovations and contributions of this paper are to improve the detection accuracy of the model, and whether separate formulation is necessary.
2. Principle of YOLOv8 detection algorithm
Figure 1 is presented in a way that feels arbitrary, with vague color distinctions, dashed boxes that are not closed, and obvious errors, so it is recommended that the figure be carefully redrawn.
3. GR lightweight model
There are some obvious formatting problems in the text, such as formulas (1) (2) (3) in the paper are recommended to be prepared with a format editor; Figure 2 out of the figure font is not uniform, it is recommended to standardize the figure.
4. Wise-IoU loss function
The formulas in the text are recommended to be prepared with a format editor, and Figure 3 is ambiguous, and it is recommended to standardize the figure.
5. Experiments and analysis of results
(1) Table 1 in the text labels the dataset with 12 categories of fault types, with no indication of what the number of each category is;
(2) Formulas are recommended to be prepared with a format editor;
(3) Figure 4 is not standardized and it is recommended that it be redrawn;
(4) Figure 6 is not standardized and it is recommended that it be redrawn;
(5) Only the labeled 5 categories were used in the actual test in Fig. 7, and whether the extra labeling was necessary since it was not used in the later experiments;
(6) In Figure 7, the first set of data from the left only tested 2 labeled fault categories, called the appropriateness of multiple types; the second group with the fourth group of samples defective features, zoomed in on the human eye is almost impossible to see, the text said that the algorithm did not miss the detection, but how to ensure that small samples labeled, trained high-precision algorithmic models, will not produce overfitting of new samples of detection data resulting in a miscarriage of justice. To consider the rigor of logic.
Reviewer 4 Report
Comments and Suggestions for Authors
The manuscript, titled "Multi-fault target detection method for transmission lines based on GR-YOLOv8," presents an approach that aims to enhance the performance of fault detection algorithms using the YOLOv8 framework. However, despite the relevance of the topic and the potential contributions to real-time fault detection in power transmission systems, the manuscript is riddled with several critical issues that must be addressed before it can be considered for publication.
The writing quality and clarity of the manuscript are particularly concerning. Many technical terms and acronyms are introduced throughout the text without proper definitions or explanations, making the content difficult to follow, even for readers familiar with the field. For example, on page 1, line 3, the acronym "YOLO" is used without clarifying what it stands for. Likewise, on lines 13 and 14, terms like "Rep vision transformer" and "Neck part" are presented with no context or explanation, resulting in a nearly incomprehensible sentence. The manuscript would benefit significantly from clear and concise explanations of all technical terms upon their first mention to improve accessibility for readers unfamiliar with the specific jargon.
In addition to issues with clarity, several instances of redundancy detract from the flow of the text. On lines 42-43, the manuscript redundantly refers to "R-CNN and regional convolutional neural networks," even though "R-CNN" is an abbreviation for Region-based Convolutional Neural Networks. This redundancy is unnecessary and adds confusion. Such issues appear multiple times throughout the manuscript and should be corrected to streamline the text.
The sentence structure in many sections also lacks coherence. For instance, on lines 13 and 14 of page 1, the phrase "Rep vision transformer enables the model to learn the global representation" is convoluted and unclear. The authors must clarify what this module does and how it contributes to the model's overall performance. Furthermore, the logic within certain sentences is weak, as demonstrated in lines 42-43, where the text mentions the advantages of deep learning-based object detection but then abruptly shifts to discussing two-stage versus one-stage detection algorithms without fully explaining how the two points relate. This inconsistency in narrative needs to be addressed for better comprehension.
Beyond the writing issues, the manuscript suffers from a lack of scientific rigor and structure in its technical sections. On page 7, section 5.2, the manuscript lists the hyperparameters used in the training process, such as the learning rate and momentum, without providing any justification for these choices or explaining their meaning. The authors must explain why these specific values were chosen and how they were tuned for optimal performance. Without this explanation, the reader cannot assess the robustness of the methodology.
Figures and equations in the manuscript are also problematic. Figure 2, found on page 4, is of low resolution and is difficult to read, while the equations in the figure are poorly formatted. These visual elements are crucial for understanding the architecture of the proposed model, and their current quality severely hinders this understanding. Furthermore, the terms "DSC" and "SC" are introduced on lines 129-132 without any prior explanation, leaving readers unfamiliar with these terms in the dark. All figures and equations must be presented accordingly, with proper explanations for each introduced term, to ensure that the manuscript is scientifically sound and comprehensible.
The experimental design, particularly the ablation studies presented in Table 2 (page 8), is also insufficiently explained. While the results show improvements, the text provides little insight into why these improvements occur. The manuscript simply presents the data without discussing the reasons behind the performance changes. Furthermore, there is inconsistency in the use of terms such as "mAP" and "average accuracy," which are used interchangeably despite representing different metrics. Precision and accuracy are distinct concepts, and the manuscript must maintain consistency in the terminology to avoid confusion. The evaluation metrics should be clearly defined and explained to ensure that the reader can follow the experimental results.
The novelty of the manuscript is another area where improvements are needed. While the use of the Rep ViT and GSConv modules is mentioned as innovative, these techniques have already been explored in other recent works. For instance, several studies have utilized lightweight convolutional modules and attention mechanisms in the context of YOLOv8 for insulator detection. The manuscript does not sufficiently differentiate its approach from these existing methods. A more thorough discussion of how the proposed method advances the state of the art is necessary to justify the claims of novelty. Additionally, the literature review is inadequate, failing to discuss relevant works in sufficient depth. A more comprehensive review of current research in lightweight fault detection models, such as recent developments in YOLOv7 and YOLOv8, would provide a stronger foundation for the manuscript's contributions.
Reproducibility is a fundamental aspect of scientific research, and the manuscript does not meet the required standards in this regard. The dataset used in the experiments, consisting of 3,600 images from the State Grid inspection, is briefly mentioned, but no details are provided regarding how these images were labeled or processed. Additionally, there is no mention of where the dataset can be accessed or how other researchers could replicate the experiments. Without this information, it is impossible to verify the results or apply the proposed method in other contexts. The authors must provide more detailed information about the dataset and ensure that the necessary resources are available for reproducibility.
The use of the Wise-IoU loss function, introduced on page 5, also lacks sufficient justification. Although the manuscript claims that this function reduces harmful gradients, there is little experimental evidence presented to support this assertion. The authors should provide a detailed comparison between Wise-IoU and other commonly used loss functions, such as GIoU or DIoU, to demonstrate the advantages of their approach. Without this comparative analysis, the choice of loss function appears arbitrary and unsupported.
In conclusion, while the manuscript addresses an important issue in the field of transmission line fault detection, it suffers from significant shortcomings in terms of writing clarity, technical rigor, and reproducibility. The lack of clear explanations, unjustified technical choices, and vague experimental results make the manuscript difficult to evaluate in its current state. Based on these substantial issues, the recommendation is "Reject". The manuscript contains serious flaws, including incomplete experimental design, a lack of necessary comparisons with current methods, and a failure to provide sufficient data for reproducibility. The authors will need to address these issues through additional experiments, clearer articulation of technical methods, and a more thorough review of the literature before the paper can be reconsidered for publication.
The quality of the English language in the manuscript is poor and requires significant improvement. Many sentences are unclear due to awkward phrasing, incorrect grammar, and a lack of coherent structure. There are frequent issues with subject-verb agreement, and technical terms are often introduced without proper explanation. For example, on lines 13-14 (page 1), the phrase "Rep vision transformer enables the model to learn the global representation" is difficult to comprehend due to its vague construction. Grammatical errors appear throughout the manuscript. For instance, the use of definite and indefinite articles is inconsistent, and punctuation is misapplied in several places, affecting the readability and flow of the text. Moreover, some sections, especially those that describe the technical methodology (like the introduction of hyperparameters), lack clarity and precision, leaving readers to guess the intended meaning. A professional copy editor should be consulted to correct these issues and improve the overall quality of the manuscript's language. This will ensure that the technical content is conveyed clearly and effectively.
Round 2
Reviewer 1 Report
Comments and Suggestions for Authors
Your revised paper presents significant improvements, particularly in addressing many of the previous concerns. Here are the additional comments based on your revision:
The problem statement is much clearer now. The distinction between GR-YOLOv8 and existing YOLO variants is well-articulated. The specific limitations of traditional algorithms (e.g., slow detection speeds and low precision) and the advantages of GR-YOLOv8 are well-explained. The introduction of RepViT and GSConv to address computation bottlenecks and the use of Wise-IoU for improving detection accuracy shows a clear technical contribution. However, further highlighting the novelty compared to previous YOLO-based transformer approaches could strengthen this.
The addition of more detailed results from ablation experiments and comparisons with other models (Table 3) is excellent. It helps to clearly show the incremental benefit of each component (GSConv, RepViT, Wise-IoU) on GR-YOLOv8's performance. However, the inclusion of more visual examples comparing GR-YOLOv8’s results with other models would make the results more accessible, particularly for those less familiar with the underlying metrics.
While you’ve mentioned real-world application scenarios, particularly in UAV-based inspections, expanding this section with more emphasis on deployment feasibility in various environmental conditions (lighting, weather, etc.) could be beneficial. Additionally, discussing potential limitations of the model (e.g., computational costs in extreme edge cases) would add balance and depth to the analysis.
The comparison of FPS, recall, and precision across models is helpful and shows GR-YOLOv8's superiority. However, a clearer visual representation of performance gains (perhaps using a bar chart or line plot comparing models on key metrics) could make this data more digestible.
The conclusion effectively summarizes the contributions, but reiterating specific numbers (e.g., improvements in mAP and FPS over existing algorithms) could leave a stronger impact on readers. It would also be helpful to mention the broader potential applications of your model beyond power line inspections.
Overall, the revisions are strong, and your paper addresses many of the earlier concerns. You’ve significantly improved the technical depth and clarity of your contributions. Keep fine-tuning the explanations of technical novelty and real-world applicability to make the paper even more compelling.
Reviewer 2 Report
Comments and Suggestions for Authors
accept
Reviewer 4 Report
Comments and Suggestions for Authors
The revised version of the manuscript titled "Transmission line defect target detection method based on GR-YOLOv8" shows significant improvement in clarity and scientific content, but there are still minor areas that need further attention to ensure the paper meets the necessary standards for publication. Below is a detailed review outlining the remaining issues and suggestions for improvement.
The quality of writing has improved, but there are still some clarity issues that need to be addressed. One recurring problem is the incorrect presentation of acronyms. In several instances, such as with "R-CNN" in lines 63-64, "SSD" on line 76, and "FPN" on line 64, the acronym is introduced before the full term. The correct format is to present the full term first, followed by the acronym in parentheses. For example, it should read "Region-based Convolutional Neural Networks (R-CNN)" rather than "R-CNN (Region-based Convolutional Neural Networks)." This is a common issue throughout the manuscript and needs to be revised consistently to ensure clarity for readers. This issue has been handled correctly in some sections, as in lines 216-217, where "Feature Pyramid Networks (FPN)" is presented in the correct order. Ensuring consistency in the presentation of acronyms will improve readability and professionalism.
Some issues with sentence structure affect the clarity of the manuscript. In lines 333-334, the sentence describing different metrics for the YOLOv8 model is unclear: "the mean of average precision and detection rate of YOLOv8 were 9.2, 0.864 and 0.882, and 68.7 FPS, respectively." This sentence mixes multiple metrics (mAP, recall, and FPS) confusingly. A clearer phrasing would separate the metrics more logically, for example: "The mean average precision (mAP) of YOLOv8 was 0.882, the recall was 0.864, and the FPS was 68.7." Correcting such sentences will make the data presentation clearer and easier to follow.
Another issue is the section numbering. Currently, the manuscript begins the introduction with the number "0," which is unconventional in academic writing. The numbering should begin with "1" for the introduction, followed by "2" for subsequent sections. This small formatting detail should be corrected to maintain the professionalism and standard structure of the manuscript.
There are also small formatting issues in the abstract, specifically related to spacing. For example, in line 12, there is extra space before the parentheses in "(You Only Look Once version 8)." Similar spacing issues can be found in other parts of the manuscript. These small errors detract from the overall presentation and should be corrected for consistency.
In conclusion, the manuscript is approaching a publishable standard but still requires some minor revisions. Once these minor revisions are addressed, the manuscript will be ready for publication.
The quality of English in the manuscript is generally good, but some areas require improvement. Minor formatting issues, such as inconsistent spacing around parentheses, must be reviewed. A careful review of these aspects will enhance the overall readability and professionalism of the manuscript.